# Beyond Parameters: Exploring Virtual Logic Depth for Scaling Laws

## Abstract

Scaling the size of large language models typically involves 3 dimensions: depth, width, and the number of parameters. In this work, we explore a 4th dimension: **virtual logical depth** (VLD), which allows increasing the effective algorithmic depth without changing the overall parameter count by reusing parameters within the model. While parameter reuse is not new, its role in scaling dynamics has remained underexplored. Unlike currently trending test-time methods, which mainly scale in token-wise, VLD alters the internal computation graph scaling during training, inference, or combination. We carefully design controlled experiments and have the following key insights on VLD scaling: 1. *Knowledge capacity vs. parameters.* At a fixed parameter count, VLD leaves knowledge capacity nearly unchanged (with only minor variance), while across models *knowledge capacity scales with the number of parameters*; 2. *Reasoning vs. reuse.* Properly implemented VLD substantially improves *reasoning ability without* increasing parameter count, decoupling reasoning from sheer model size. This provides a **new possibility for scaling** besides the current token-wise test-time scaling used by most reasoning models. 3. *Robustness and generality.* The trend of improved reasoning persist across architectures and configurations (e.g., different reuse schedules and step counts), indicating that VLD captures a general scaling behavior. These findings not only provide useful insights into the future model scaling strategies, but also introduce an even deeper question: Does super intelligence necessarily require ever-larger models, or could it have some trade-offs by re-using parameters and increasing virtual logic depth? We believe that there are many unknown dynamics within the model scaling that need exploration. Codes are available at https://anonymous.4open.science/r/virtual_logical_depth-8024/.

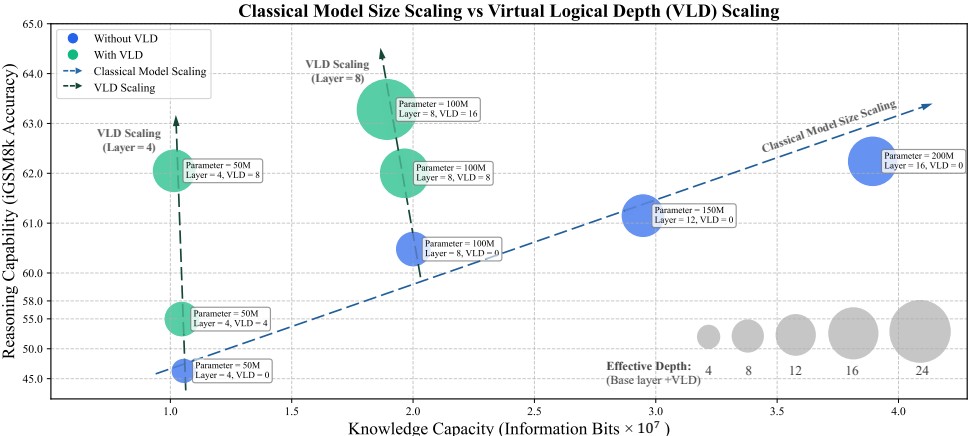

Figure 1: **Comparison between classical model size scaling vs. VLD scaling.** Bubble size proportional to total effective depth (base layer depth + virtual logical depth). Blue bubbles represent standard models without VLD scaling but following the classical model size scaling, while green bubbles show models with VLD scaling applied, demonstrating near-vertical scaling paths. VLD scaling significantly enhances reasoning capability (y-axis) while keeping the knowledge capacity (x-axis) almost constant without significant variations.

# 1 INTRODUCTION

Scaling up large language models (LLMs) has long been regarded as a primary driver of their rapid progress (Kaplan et al., 2020; Hoffmann et al., 2022; Brown et al., 2020; Wei et al., 2022). Scaling typically proceeds along multiple axes, including model size, data, compute (Wei et al., 2022), and more recently inference-time scaling (Snell et al., 2024; Jaech et al., 2024). The model size scaling direction typically occurs along three dimensions: depth, width, and the total number of parameters [1]. Contemporary LLMs often exceed 100B parameters (Wei et al., 2022), enabling them to store vast knowledge. However, parameter growth does not straightforwardly translate into stronger reasoning capabilities.

In contrast, human cognition demonstrates the opposite profile. Memory is relatively limited—hence the invention of external storage such as writing—yet reasoning and abstract problem solving are remarkably strong (Chollet, 2019; Xu & Poo, 2023; Wang & Li, 2024), largely supported by specialized circuits in the prefrontal cortex (Raichle, 2009). This asymmetry highlights a fundamental tension: while current scaling paradigms expand memory-like capacity, they do not necessarily improve reasoning, which is arguably central to achieving super-intelligence. Ideally, an intelligent system would combine (i) a powerful reasoning core, (ii) a compact but essential internal memory, and (iii) external retrieval for domain knowledge (Kumaran et al., 2016).

Motivated by this perspective, we introduce a new scaling dimension that *deliberately limits knowledge capacity in order to encourage improvements in reasoning*. Specifically, we **revisit weight sharing** in the transformer architecture and show that repeating layers with tied parameters yield surprisingly strong benefits. We term the resulting phenomenon **virtual logical depth** (VLD). While related techniques—such as parameter reuse, layer repetition, and equilibrium models (Lan et al., 2020; Dehghani et al., 2019; Bai et al., 2019; Hay & Wolf, 2024; Bhojanapalli et al., 2021; Reid et al., 2021; Takase & Kiyono, 2021; Liu et al., 2023)—have been explored in other contexts, their scaling dynamics have remained underexplored. A closely related effort, looped transformers (Saunshi et al., 2025), studies reasoning capacity, but our focus is on *scaling laws*: we conduct systematic experiments and uncover new properties of VLD as a scaling strategy.

Through carefully controlled experiments, we investigated the characteristics of VLD scaling, specifically measuring reasoning capabilities and knowledge capacity under various configurations. We discovered that VLD scaling shows very different properties from classical model size scaling. In brief, VLD scaling *forces the knowledge capacity of the model to stay almost constant (with non-significant variations), while significantly improving the reasoning capability*. We illustrate a part of our main findings in Figure 1, which compares the two scaling strategies. These findings hold for various model and experiment configurations, and are likely to be universal under our scope of experiments. Under reasonable conditions, *we don't have to increase the number of parameters in order to increase the inherent reasoning capability of the model*.

Beyond the scope of this work, we would like to raise a even deeper question: *Do we really need a lot of parameters to achieve super-intelligence?* Of course this first depends on how we define super-intelligence. Humans have poor memory but good reasoning capability and agency capability (e.g., subjective initiatives). *Under limited human brain capacity, why did the natural evolution chose to allocate more capacity on reasoning capability and agency capability, but only some essential capacity for memory?* It is not impossible that under a small but essential amount of parameters and memory capacity, we will be able to push the reasoning capability and agency capability to the extreme and achieve super-intelligence in the far future. We would like to move a small step towards this ultimate goal, and open-source all source code of our current work to facilitate future research.

# 2 RELATED WORK

**Parameter Sharing and Layer Reuse.** Parameter reuse has garnered considerable attention in recent research. (Reid et al., 2021) introduced Subformer, which reused 50% of parameters for compression purposes, while (Liu et al., 2023) presented ESP (Efficient Shared Parameterization)

---

[1]Since depth and width directly influence the number of parameters, these three dimensions are not entirely independent. However, the total parameter count can indeed be adjusted independently by modifying the inner dimension of the MLP layer within a transformer block.

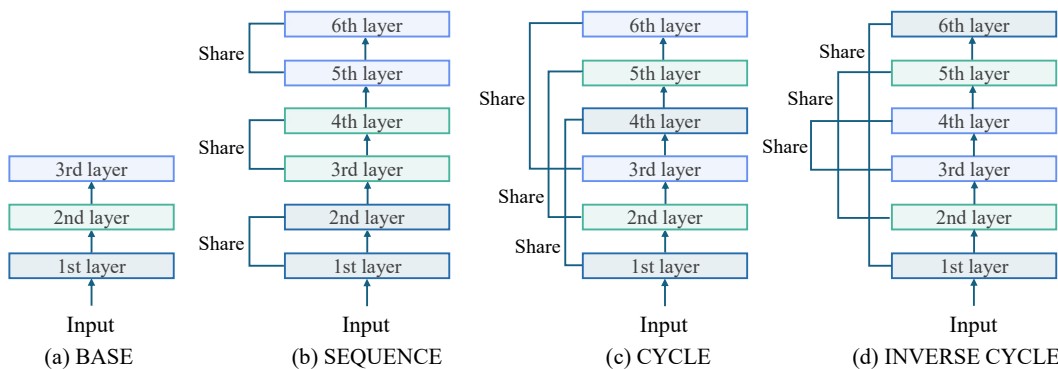

(a) BASE  (b) SEQUENCE  (c) CYCLE  (d) INVERSE CYCLE

Figure 2: **Different patterns of parameter reuse to increase VLD while keeping the total number of parameters constant.** (a) an example of a standard transformer with 3 layers. (b) sequentially repeat neighboring layers. (c) cycle-repeat the layers. (d) repeat layers in an inverse-cycle order. Under the same pattern, two layers with the same color share the same parameters. In all cases, the actual number of parameters do not change.

that reused central tensor matrices. However, both introduced atypical elements to transformer architectures, complicating direct comparison with standard models. (Bhojanapalli et al., 2021) developed Reuse Transformers, exploring attention mechanism reuse while sharing less than 10% of attention heads. (Takase & Kiyono, 2021) proposed three distinct layer-sharing patterns but conducted experiments only on small networks without scaling analysis. More recently, (Hay & Wolf, 2024) proposed Dynamic Layer Tying, employing reinforcement learning to identify optimal layer-sharing configurations. In parallel, several studies have explored Looped Transformers (Fan et al., 2024; Saunshi et al., 2025; Yang et al., 2023), which iteratively apply transformer layers to enhance model capabilities without proportionally increasing parameters. Our work builds upon these foundations by systematically analyzing VLD scaling dynamics and its distinct impact on reasoning capability versus knowledge capacity.

**Measuring Reasoning and Knowledge Capacity.** Quantitative assessment of reasoning and knowledge remains challenging. For reasoning capabilities, (Ye et al., 2024) employed synthetic GSM8K benchmarks to isolate inherent reasoning abilities from memorized solutions. Regarding knowledge capacity, (Allen-Zhu & Li, 2024) demonstrated that LLMs store approximately 2 bits of knowledge per parameter, providing a quantitative metric for model memory. (Carlini et al., 2022) established methodologies for quantifying memorization in LLMs, offering insights into information retention mechanisms. Understanding how large language models (LLMs) store and retrieve information is central to evaluating their memory mechanisms. (Liang et al., 2024) introduced the SAGE framework, which enhances LLM agents with reflective and memory-augmented abilities. (Xu et al., 2025) propose A-Mem, an agentic memory framework designed to enhance the long-term reasoning and adaptability of LLM agents.

## 3 VIRTUAL LOGICAL DEPTH

We define **Virtual Logical Depth (VLD)** as the effective algorithmic depth of a model minus the number of base layers. Figure 2 provides an illustration: here, 3 base layers are reused to form 3 VLD layers. Instead of introducing a new architecture, our goal is to establish a new *scaling dimension* within the standard transformer framework. Concretely, we retain the transformer architecture but repeat its layers with shared parameters, following the parameter reuse insights of Takase & Kiyono (2021).

We explore three reuse patterns—*sequence*, *cycle*, and *inverse cycle*—as shown in Figure 2 (b–d). In the **sequence** mode, adjacent layers share parameters. In the **cycle** mode, several layers form a block that is repeated. In the **inverse cycle** mode, the same block is repeated but in reversed order. Detailed implementation and intuition for each strategy are described in Appendix D.

**Problem Statement.** Given the VLD scaling and standard model size scaling, find the difference of their influence on knowledge capacity and reasoning capability of the model.

The above question requires us to develop reasonable approaches to measure the knowledge capacity and reasoning capability. Our high-level guideline is that such kind of measurement should be done with rigorously controlled experiments like in (Allen-Zhu & Li, 2024). We got inspiration from (Allen-Zhu & Li, 2024) and develop the measurement approaches as explained in Section 3.1 and Section 3.2.

### 3.1 Measurement of Knowledge Capacity

Knowledge capacity can estimated by measuring the maximum information entropy (Gray, 2011; Volkenstein, 2009) that a LLM absorbs. For a discrete variable $X$ that has $n$ possible values $x_1, x_2, \cdots, x_n$, the expected information entropy (in bits) for a realization of $X$ is:

$$H(X) = -\sum_{i=1}^{n} p(x_i)\log_2 p(x_i)$$

Given this, we follow these steps to measure the knowledge capacity:

**Step 1: Construct a Random Number Dataset.** We construct a random number dataset, which is a sequence of $k$ numbers, where each number is a realization of $X$, uniformly drawn from the $n$ possible values. The information entropy bits of this dataset is:

$$H_1 = -k\sum_{i=1}^{n} \frac{1}{n}\log_2 \frac{1}{n} = k\log_2 n$$

**Step 2: Train a LLM to Remember the Random Number Sequence.** Now assuming we train a LLM to fit the dataset. Each random number corresponds to exactly one token. The softmax output of the LLM is a length-$n$ vector that predicts the probability of the next token. If the LLM is not trained at all, the softmax vector could have uniform probability output. But if the LLM is trained to remember the sequence, it should output a higher probability for the correct next token. To be more mathematically specific, we first ensure that each $x_i$ only occupy one token position in the tokenizer, then we build the training set by putting the $k$ realization of $X$ into a fixed-order sequence. After this step, we already obtained a sequence of length $k$ where each token has uniform distribution on $n$ possible values. Then a LLM is trained to fit the sequence.

**Step 3: Calculate Knowledge Capacity from Softmax Output.** First notice that this experiment only relies on a training set but not a validation set, because the objective is to let the LLM remember the training set as good as possible. Once the training is converged, let the softmax probability output at the $j^{th}$ input for the $i^{th}$ value of $X$ is $p_j(x_i)$. To be clear, $p_j(x_i)$ is the probability that the $(j+1)^{th}$ number in the sequence is $x_i$, predicted by the LLM. The LLM is expected to output a higher probability if that number is really $x_i$ and lower probability otherwise. If the LLM remembers everything and has absolute confidence, then $p_j(x_i)$ is 1 for the correct $x_i$ and is 0 otherwise. But in reality the LLM has limited knowledge capacity. We define quantity $H_2$ as:

$$H_2 = -\sum_{j=1}^{k}\sum_{i=1}^{n} p_j(x_i)\log_2 p_j(x_i)$$

If the LLM remembers everything in the dataset, then $H_2$ should be close to zero [2]. If the LLM remembers nothing, then $H_2$ should be $H_1$. The difference between $H_1$ and $H_2$ is:

$$\Delta H = H_1 - H_2$$

The $\Delta H$ is essentially the information entropy that the LLM absorbed, which can be viewed as a form of knowledge capacity. In the case that $H_1$ is larger than the knowledge capacity of the LLM,

---

[2] There are some boundary conditions to be satisfied before we can make such claim. But as the sequence length becomes extremely long, such as $640k$ in our experiments, the influence of the boundary conditions is averaged out and does not impact the conclusion of our experiments.

$H_2$ cannot reach 0 because there is always a part of the information entropy that the LLM cannot absorb. In this case, the difference $\Delta H$ effectively reflects the maximum knowledge capacity of the LLM being trained. We will present more details on how we configured the experiments to measure the knowledge capacity in Section 4 the experiments.

## 3.2 Measurement of Reasoning Capability

Developing a rigorous measurement of reasoning capability is harder than measuring the knowledge capacity. Reasoning capability does not have a clear unit like information entropy bits as we have in knowledge capacity. Therefore, we adopt widely used empirical metrics as measurements, including accuracy on mathematical problems, scientific reasoning tasks, and code generation benchmarks. To ensure comprehensive evaluation, we employ both synthetic and real-world datasets to assess VLD's impact on reasoning capabilities.

### 3.2.1 Synthetic Data

**Step 1: Synthesize Large Math Datasets.** Inspired by (Ye et al., 2024), we synthesize the highly diverse iGSM (Ye et al., 2024; Cobbe et al., 2021) dataset to measure the mathematical reasoning capability of LLMs and study how virtual logical depth influences such capability. In each problem of the iGSM dataset, the problem statement implies the dependencies of a set of variables. Given the value of some variables, the goal is to infer the value of a target variable. The problem difficulty is measured by the number of algebra **operations** to derive the target value. Since the dataset can theoretically contain more than 90 trillion (Ye et al., 2024) solution templates, which is much larger than the number of parameters in LLMs, LLMs cannot solve the problems by simply memorizing the solution templates. The process of synthesizing the iGSM dataset is thoroughly described in (Ye et al., 2024) and will not be covered in this manuscript. One example of the synthesized problem is provided below, with additional training data examples included in Appendix E.1.

---

**Question:** Question: The number of each Lungs's Platelets equals each Lungs's B Cells. The number of each Pleural Cavity's Platelets equals 0. The number of each Pleural Cavity's B Cells equals 11 more than the sum of each Lungs's Platelets and each Lungs's B Cells. The number of each Lungs's B Cells equals 10. How many B Cells does Pleural Cavity have? **Solution:** Define Lungs's B Cells as p; so p = 10. Define Lungs's Platelets as r; so r = p = 10. Define Pleural Cavity's B Cells as u; m = r + p = 10 + 10 = 20; so u = 11 + m = 11 + 20 = 8. **Answer:** 8. [a].

---

[a]In order to control the complexity of pure numerical calculations, all solutions are mod 23. In this example, 11 + 20 := (11 + 20) mod 23 = 8.

---

**Step 2: Train LLM on Synthetic Math Data and Evaluate the Accuracy.** Solving these questions basically requires the LLM to produce operations step-by-step to derive the value of the target variables. The number of steps is a measurement of the problem's complexity. We generated a non-overlapping training set and validation set. The training set contains problems with operations no greater than 15. The validation set contains problems with operations equal to 15, which represents the highest difficulty that the model saw during training. During validation, we feed the problem statement into the LLM and let it generate the solution and final answer. The detailed experiment configurations are described in Section 4.

### 3.2.2 Real-World Data Evaluation

**Step 1: Construct Multi-Domain Training Corpus.** To validate VLD's effectiveness beyond synthetic scenarios, we construct a comprehensive real-world training corpus spanning multiple reasoning domains. This corpus comprises diverse datasets organized by category as shown in Table 3 in the Appendix E.3. This multi-domain approach ensures the model is exposed to diverse reasoning patterns including conversational AI, instruction following, programming tasks, mathematical problem-solving, and scientific reasoning across over 2.3 billion tokens.

**Step 2: Evaluate on Real-World Reasoning Benchmarks.** We assess reasoning capability using established benchmarks that evaluate distinct aspects of reasoning. For **mathematical reasoning**,

we use Math500(Hendrycks et al., 2021) and AIME(Mathematical Association of America) to evaluate multi-step mathematical problem solving capabilities. For **scientific knowledge**, we employ GPQA(Rein et al., 2024) to assess graduate-level scientific reasoning and factual knowledge. For **code generation**, we utilize HumanEval(Chen et al., 2021) and MBPP(Austin et al., 2021) to measure functional programming and algorithmic reasoning abilities. These benchmarks provide comprehensive coverage of reasoning modalities, enabling assessment of VLD's impact across mathematical, scientific, and computational domains. The detailed experiment configurations are described in Section 4.

## 4 EXPERIMENTS

Our experiments are designed to systematically investigate the distinct impacts of *Virtual Layer Depth (VLD)* scaling on two fundamental aspects of Large Language Models (LLMs): **knowledge capacity** and **reasoning capability**. We aim to understand how VLD patterns influence these attributes compared to classical model scaling approaches. Our experiments are centered around two main questions:

- How does VLD affect knowledge capacity and reasoning capability compared to classical parameter scaling?
- Do the observed effects of VLD persist across both pretraining and fine-tuning regimes?

**Shared evaluation protocol.** Unless noted otherwise, we align tokenizer/vocabulary, per-layer parameterization, optimizer, schedule, batch size, and decoding settings within each table to isolate the effect of VLD. We denote VLD *Depth* $\times k$ as repeating each virtual block $k$ times according to the pattern definitions in Section 3. Reporting is consistent across rows; any deviation is explicitly stated.

### 4.1 KNOWLEDGE CAPACITY EXPERIMENTS

**Random Sequence Data Setup.** For the knowledge capacity experiment, we construct a high-entropy random number corpus parameterized by $(n, k)$ as in Section 3.1. The $n$ is chosen based on the default vocabulary size of our transformer training pipeline and does not have a specific meaning; we use $n = 50257$. The $k$ is chosen to be sufficiently large so that the dataset contains enough information entropy. Prior research (Allen-Zhu & Li, 2024) established that approximately 2 bits of information can be stored per model parameter. With our smallest model containing 5M parameters, we designed experiments to store approximately 10M bits of information (aligning with the 2 bits per parameter capacity), requiring a sequence length of approximately $k = 640,000$ random tokens. Training data examples consist of IID token IDs generated from $[0, 50256]$.

**Models & Metric.** We established four 4-layer GPT-2 models with increasing parameter counts (5M, 10M, 15M, and 20M). The layer size we use is 92 for 5M model and 184 for 20M model, which is consistent with the settings in (Allen-Zhu & Li, 2024). To investigate VLD effects while holding parameters fixed, we implemented the three VLD patterns described in Section 3 on the smallest (5M) and largest (20M) baseline models. For each pattern, we studied repetition factors of $\times \{1, 2, 3\}$. After training, we input the training sequence to the LLM and computed the sum of entropy of the model's softmax outputs, then compared the total entropy of the dataset with this measured value. The difference, $\Delta H$ (Section 3.1), represents the amount of information successfully absorbed by the model. The mean absorbed information entropy is defined as the average amount of information absorbed per token, calculated as the reduction from the original uncertainty (15.6 bits for $n = 50257$) to what the model predicts. We used a maximum sequence length of 768 during training and 1024 during evaluation. The detailed model configuration is listed in the Appendix F.1.

### 4.2 REASONING CAPABILITY EXPERIMENTS

To ensure comprehensive evaluation and robust validation of VLD's effectiveness, we conduct reasoning capability experiments in two settings: (1) Pretraining with synthetic data for controlled analysis, (2) Post-training with diverse real-world benchmarks for practical validation.

Table 1: GPT-2 performance on synthetic GSM-8K tasks across multiple VLD patterns and depths. Accuracy (%) at controlled operation counts. "VLD Depth" denotes the repetition factor applied to the base layer count. Cell shading encodes accuracy (darker = higher); within each VLD pattern, accuracy generally increases with VLD depth, though not strictly monotonically

| Model | VLD Pattern | VLD Depth* | iGSM Acc. (4 Base Layer) | | | iGSM Acc. (8 Base Layer) |
|---|---|---|---|---|---|---|
| | | | op15 | op20 | op21 | op15 |
| GPT-2 | Base | – | 46.3 | 21.8 | 21.2 | 60.5 |
| | Cycle | ×1 | 54.9 | 26.4 | 26.2 | 62.0 |
| | | ×2 | 62.1 | 30.4 | 35.4 | 63.3 |
| | | ×3 | 61.6 | 30.8 | 32.0 | 61.0 |
| | | ×4 | 65.7 | 39.2 | 33.0 | 66.8 |
| | | ×5 | 70.7 | 43.8 | 40.2 | – |
| | Sequence | ×1 | 50.7 | 25.6 | 21.2 | 59.2 |
| | | ×2 | 51.2 | 27.0 | 22.2 | 62.1 |
| | | ×3 | 55.5 | 28.0 | 25.0 | 62.9 |
| | Inv. Cycle | ×1 | 43.9 | 17.4 | 15.8 | 53.3 |
| | | ×2 | 50.8 | 24.2 | 18.2 | 62.0 |
| | | ×3 | 54.8 | 25.2 | 22.2 | 62.8 |

**Notes.** (i) "op$X$" denotes test problems with exactly $X$ operations; (ii) rows within each backbone share identical training and decoding settings; (iii) reported numbers are the mean test accuracy over 2–3 checkpoints after training convergence.
\* VLD Depth indicates the multiplication factor applied to the base model's layer count. Models all train from scratch.

Table 2: LLaMA-3.2-3B-Instruct on real-world benchmarks over Base model vs. Cycle-VLD. Performance (%) under fixed prompts/decoding per benchmark. Both models are fine-tuned on our SFT corpus from the same pretrained weights.

| Model | VLD Pattern | Real-World Benchmarks Performance (%) | | | | |
|---|---|---|---|---|---|---|
| | | Math500 (Acc.) | GPQA (Acc.) | AIME (Acc.) | HumanEval (pass@1) | MBPP (pass@1) |
| **Llama-3.2-3B-Inst** | Base | 30.40 | 29.80 | 3.33 | 37.79 | 38.36 |
| | Cycle | 35.40 | 32.32 | 6.67 | 39.52 | 40.22 |

### 4.2.1 PRETRAINING SETTING

**Synthetic Data & Models Setup.** For the reasoning capability experiment, we build dataset generated from the iGSM framework (Ye et al., 2024), comprising 500K mathematical problem samples with carefully controlled difficulty parameters. Each problem adheres to consistent complexity constraints with a maximum of 15 operational steps, ensuring reliable measurement of reasoning abilities. We generated non-overlapping training and validation sets, with training problems having ≤15 operations and validation problems having exactly 15 operations. We constructed four GPT-2 based models with identical per-layer parameter counts but varying native layer depths (4, 8, 12, and 16 layers, corresponding to roughly 50M, 100M, 150M, and 200M parameters). We applied the three VLD patterns to the 4-layer and 8-layer models with multiplication factors of $\times\{1, 2, 3\}$ (and up to $\times 5$ for cycle pattern). All models were trained to convergence on $8\times$ A100 GPUs using Adam with learning rate $2\times 10^{-5}$. We place more details of model and training in Appendix F.2.

**In-distribution & Out-of-distribution Evaluation** To assess VLD's impact on reasoning capabilities, we employ a controlled evaluation protocol measuring exact-answer accuracy across different complexity levels. Our evaluation encompasses both in-distribution and out-of-distribution scenarios to examine reasoning robustness under controlled shifts in problem difficulty. For *In-Distribution evaluation*, we test models on 500 synthetic iGSM samples with exactly 15 operations, matching the maximum complexity seen during training. Each problem requires multi-step mathematical reasoning, and we calculate exact-answer accuracy by comparing model-generated solutions against ground truth labels using string matching on the final numerical answers. To evaluate robustness under *Distribution Shifts*, we assess model performance on higher complexity problems containing 20 and 21 operations, respectively. These out-of-distribution problems maintain the same mathematical reasoning structure

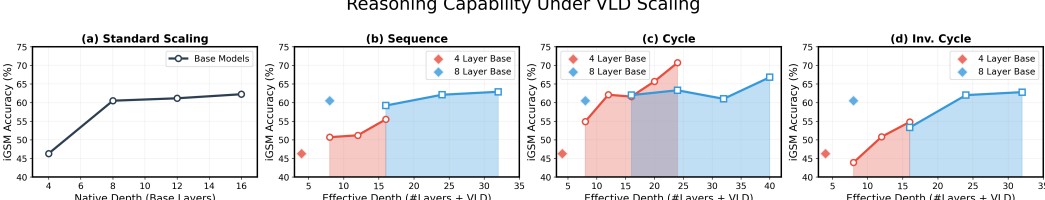

Figure 3: **Reasoning Capability Under VLD patterns.** (a) Standard scaling with native depth (16-layer $\approx$ 200M; 12-layer $\approx$ 150M). (b–d) Sequence, Cycle, and Inverse Cycle applied to 4-layer (50M; red) and 8-layer (100M; blue) backbones. Train from scratch, test with op15 GSM data.

but require additional computational steps, allowing us to measure whether VLD improvements generalize beyond the training distribution. All evaluations use identical decoding parameters (greedy decoding) and prompting strategies across model variants to ensure fair comparison. Models generate complete step-by-step solutions, from which we extract final answers for accuracy assessment.

### 4.2.2 Post-Training Setting

**Training Data Composing & Models Setup.** We construct a comprehensive multi-domain SFT corpus for fine-tuning the LLaMA-3.2-3B-Instruct model, totally 2.3 billion tokens. Following the methodology established in Shen et al. (2024), we carefully curate datasets across three critical reasoning domains with strategic subset selection to ensure balanced coverage: (1) For natural language understanding and instruction following, we incorporate xP3x (Muennighoff et al., 2022), OpenAssistant (LAION-AI, 2023) , OpenHermes (Teknium, 2023) , and UltraChat (Ding et al., 2023). (2) For code generation capabilities, we include Magicoder-OSS (Wei et al., 2023) , Magicoder-Evol (Wei et al., 2023) , Code-290k-ShareGPT, CommitPackFT (Muennighoff et al., 2023), and Evol-Code Alpaca (Luo et al., 2023) . (3) For mathematical reasoning, we utilize Open-web-math (Paster et al., 2023) , algebraic-stack (Azerbayev et al., 2023) , TemplateGSM (Fu et al., 2023) , StackMathQA (Zhang, 2024) , and OpenR1-Math-220k (Open R1 Team, 2025). This diverse composition ensures comprehensive exposure to varied reasoning patterns and generation challenges across multiple modalities.

For model training, we fine-tune both Base and Cycle VLD variants of LLaMA-3.2-3B-Instruct on the multi-domain corpus using LoRA(Hu et al., 2022) with standard supervised fine-tuning procedures. Both models are initialized with identical pretrained weights to ensure fair comparison, with the Cycle VLD variant incorporating the layer repetition pattern before training commences. We place detailed training configurations, dataset statistics and samples in Appendix F.3.

**Test Datasets & Metrics & Model Configuration.** We conduct comprehensive evaluations on diverse real-world reasoning benchmarks following established evaluation protocols (Habib et al., 2023). The evaluation encompasses three critical domains: (1) mathematical reasoning datasets: Math500 (Hendrycks et al., 2021) and AIME (Mathematical Association of America); (2) scientific reasoning dataset: GPQA (Rein et al., 2024); (3) code generation datasets: HumanEval (Chen et al., 2021) and MBPP (Austin et al., 2021). For evaluation metrics, we maintain consistency with prior work (Habib et al., 2023; Hendrycks et al., 2021; Rein et al., 2024), employing exact match accuracy for mathematical and scientific reasoning tasks (Math500, AIME, GPQA) and pass@1 accuracy for code generation benchmarks (HumanEval, MBPP) to measure functional correctness. We conduct experiments comparing the LLaMA-3.2-3B-Instruct Base model against the Cycle VLD variant. We maintain consistent hyperparameters, optimizer settings, and evaluation protocols across both configurations to ensure fair comparison. More testing details are provided in Appendix E.3.

### 4.3 Experiment Results and Findings

**Primary Finding.** VLD scaling significantly increases reasoning capability while maintaining knowledge capacity nearly constant. As demonstrated in Table 1 and Figure 3, applying VLD patterns dramatically boosts models' reasoning performance across all tested configurations. Most notably, the cycle pattern achieves substantial improvements, increasing the 4-layer model's accuracy from

46.3% to 70.7% with a VLD factor of 5. Simultaneously, Figure 4 confirms that despite these reasoning enhancements, knowledge storage capacity remains largely unchanged across all VLD patterns and repetition factors. This distinctive attribute clearly differentiates VLD scaling from classical model scaling, enabling substantial reasoning improvements without significantly altering knowledge capacity.

Beyond the primary result, we identify several supplementary findings that warrant further discussion:

**Parameter-efficient reasoning improvements through VLD.** Smaller VLD-augmented models can surpass larger non-VLD baselines on multi-step reasoning (Fig 1). For instance, the 50M-parameter model using the cycle pattern (factor 2, effective depth=12) achieves 62.05% accuracy, surpassing the native 150M-parameter model (12-layer) with only 61.15% accuracy. This remarkable outcome emphasizes the efficacy of VLD as a parameter-efficient alternative for enhancing reasoning capabilities, indicating that increased reasoning does not necessitate an expansion in model parameters.

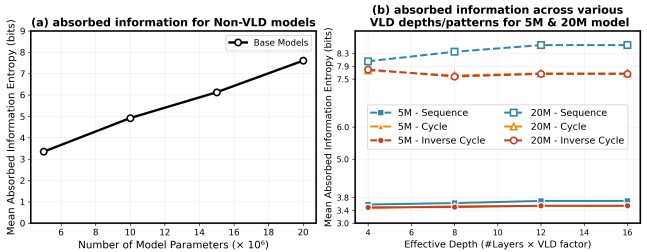

Figure 4: **Knowledge Capacity Under VLD.** (a) Non-VLD: capacity increases with parameter count. (b) At fixed parameters, absorbed information stays nearly constant across VLD depths/patterns for 5M and 20M models.

**Robustness under distribution shifts in reasoning depth.** With increasing operation counts (15/20/21 operations), cycle-VLD consistently outperforms corresponding base models across all difficulty levels (Table 1). This supports improved robustness under controlled shifts in depth-of-reasoning, with larger operation counts corresponding to more challenging mathematical problems.

**Cross-domain generalization to real-world applications.** VLD benefits transfer beyond synthetic environments to practical applications. Real-world benchmark validation using Llama-3B confirms synthetic results with consistent improvements across mathematical reasoning (Math500, AIME), scientific reasoning (GPQA), and code generation (HumanEval, MBPP) tasks (Table 2), demonstrating universal applicability.

**Non-monotonic scaling behavior with occasional performance plateaus.** While general findings indicate consistent trends, scaling exhibits occasional counterintuitive results. For example, cycle pattern shows VLD factor 3 (61.6%) performing slightly worse than factor 2 (62.1%) for the 4-layer model, and Figure 3(c) shows depth 32 yielding lower accuracy than depth 16. These anomalies suggest potential performance bottlenecks when scaling exclusively in single dimensions, consistent with prior research indicating effective scaling requires balancing across multiple dimensions (Wei et al., 2022). Future work will explore non-monotonic cases at extreme depths, test robustness under broader distribution shifts, and report multi-seed statistics.

## 5 CONCLUSION

In this study, we studied Virtual Logical Depth (VLD), a novel scaling dimension that enhances reasoning capabilities without increasing parameter counts. Our experiments demonstrate that while traditional scaling linearly increases knowledge capacity with parameter count, VLD uniquely maintains a nearly constant knowledge capacity but significantly boosts reasoning performance. Remarkably, smaller models with VLD can surpass larger standard models in complex reasoning tasks. Future research should explore combining VLD with classical scaling dimensions and investigate non-monotonic cases at extreme effective depths. Our open-source implementation is available online to facilitate further exploration and practical integration of VLD scaling.

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

APPENDIX CONTENTS

# A    THE USE OF LARGE LANGUAGE MODELS

Large Language Models (LLMs) were used as a general-purpose assist tool during the writing and revision process of this paper. Specifically, LLMs were employed to improve the clarity, coherence, and grammatical accuracy of the manuscript text. All technical content, experimental design, data analysis, and scientific conclusions presented in this work are entirely the product of the authors' original research and intellectual contributions. The authors take full responsibility for all content and claims made in this paper.

# B    ETHICS STATEMENT

This research adheres to the ICLR Code of Ethics. Our work involves computational experiments on synthetic datasets and publicly available benchmarks, with no direct human subjects involvement. All datasets used are either synthetically generated or publicly available with appropriate licensing. The experimental methodologies and applications presented pose no foreseeable harmful implications. We acknowledge our responsibility to ensure that our research contributions are used ethically and do not enable malicious applications.

# C    REPRODUCIBILITY STATEMENT

To ensure reproducibility of our results, we provide comprehensive implementation details throughout the paper and appendix. All model architectures, training hyperparameters, and evaluation protocols are specified in detail in Sections 3, Sections 4 and Appendix. The synthetic iGSM dataset generation procedures are fully described in Section 4 and Appendix E.2, with example problems provided. Real-world benchmark evaluations follow established protocols with specific details in Appendix D.3. Code for VLD pattern implementation and experimental reproduction are available in https://anonymous.4open.science/r/virtual_logical_depth-8024/.

# D    VIRTUAL LOGICAL PATTERN SELECTION

Our VLD strategies follow insights from prior work on parameter sharing across layers in Transformers. The intuition behind each strategy is:

- **SEQUENCE**: This represents the most straightforward implementation of parameter sharing, creating uniform blocks of repeated layers throughout the network.
- **CYCLE**: This maintains regularity in parameter usage patterns while ensuring each unique layer appears at consistent intervals throughout the network depth.
- **INVERSE CYCLE**: This strategy is motivated by key findings from (Liu et al., 2023) and (Takase & Kiyono, 2021), who reported that higher decoder layers tend to obtain larger gradient norms during training. Their findings imply that higher layers require more degrees of freedom than lower layers for optimal expressiveness. Therefore, by this pattern, we can reuse lower-layer parameters in higher positions while preserving the critical expressiveness.

# E    DATASET CONSTRUCTION AND SPECIFICATIONS

## E.1    RANDOM SEQUENCE DATA FOR KNOWLEDGE CAPACITY

For the knowledge capacity experiment, we construct a high-entropy random number corpus parameterized by $(n, k)$ as described in Section 3.1. The parameter $n$ is chosen based on the default vocabulary size of our transformer training pipeline ($n = 50257$). The sequence length $k$ is determined to ensure sufficient information entropy for capacity measurement. One example of the dataset can be sequence like: "43497, 6111, 32823, 47737, 46351, 1395, 2263, 48286, 13532 ...". We used a maximum sequence length of 768 during training and 1024 during evaluation.

Following prior research (Allen-Zhu & Li, 2024) which established that approximately 2 bits of information can be stored per model parameter, we design experiments to store approximately 10M

bits of information (aligning with the 2 bits per parameter capacity for our smallest 5M parameter model). This requires a sequence length of approximately $k = 640,000$ random tokens.

**Data Generation Protocol:**

- Training data examples consist of IID token IDs generated from $[0, 50256]$
- Each sequence contains exactly $k = 640,000$ tokens
- Tokens are sampled uniformly from the vocabulary to maximize entropy
- No preprocessing or filtering is applied to maintain maximum randomness

### E.2 iGSM Synthetic Dataset Construction

The iGSM synthetic dataset construction follows rigorous specifications to ensure controlled experimental conditions for reasoning capability evaluation.

**Complexity Constraints.** Each problem adheres to the following parameters: (1) Maximum of 15 operational steps (individual mathematical operations required for solution). (2) Up to 20 edges in the structural dependency graph (connections between operations). (3) Permutation level of 5 (determining how narrative elements are arranged in problem text). (4) Comprehensive solution formatting with step-by-step pathways

**Quality Control.** All problems underwent rigorous validation to ensure solution accuracy. To maintain balanced difficulty distribution, we filtered problems using solution template hash values (a metric quantifying solution pattern complexity), retaining only those with hash values below 17.

**Problem Structure.** Each sample includes the problem text, step-by-step solution pathway, and final answer, with most problems requiring multi-step logical reasoning to reach the correct conclusion.

---

**Question:** The number of each Music Room's Clear Backpack equals 20. The number of each Music Room's Musical Instrument Backpack equals each Anthropology Classroom's Toy Backpack. The number of each Photography Studio's Toy Backpack equals 21 more than the difference of each Literature Classroom's Backpack and each Literature Classroom's Diaper Backpack. The number of each Literature Classroom's Musical Instrument Backpack equals each Music Room's Backpack. The number of each Music Room's Toy Backpack equals 9 times as much as the sum of each Anthropology Classroom's Toy Backpack and each Music Room's Musical Instrument Backpack. The number of each Anthropology Classroom's Clear Backpack equals each Photography Studio's Backpack. The number of each Literature Classroom's Diaper Backpack equals 20 times as much as each Music Room's Clear Backpack. The number of each Anthropology Classroom's Toy Backpack equals 2. The number of each Photography Studio's Clear Backpack equals each Literature Classroom's Backpack. *Find the number of Anthropology Classroom's Clear Backpack.* **Solution:** Define Anthropology Classroom's Toy Backpack as C; so C = 2. Define Music Room's Musical Instrument Backpack as X; so X = C = 2. Define Music Room's Toy Backpack as M; N = C + X = 2 + 2 = 4; so M = 9 × N = 9 × 4 = 13. Define Music Room's Clear Backpack as R; so R = 20. Define Music Room's Backpack as s; A = M + X = 13 + 2 = 15; so s = A + R = 15 + 20 = 12. Define Literature Classroom's Musical Instrument Backpack as n; so n = s = 12. Define Literature Classroom's Diaper Backpack as h; so h = 20 × R = 20 × 20 = 9. Define Literature Classroom's Backpack as x; so x = n + h = 12 + 9 = 21. Define Photography Studio's Toy Backpack as t; d = x - h = 21 - 9 = 12; so t = 21 + d = 21 + 12 = 10. Define Photography Studio's Clear Backpack as q; so q = x = 21. Define Photography Studio's Backpack as r; so r = t + q = 10 + 21 = 8 [a]. Define Anthropology Classroom's Clear Backpack as F; so F = r = 8. **Answer:** 8.

---

[a] In order to control the complexity of pure numerical calculations, all solutions are mod 23. In this example, 10 + 21 := (10 + 21) mod 23 = 8.

---

### E.3 Real-World Training Corpus Specifications

For the post-training experiments, we constructed a diverse multi-domain corpus totaling approximately 2.3 billion tokens across natural language, code, and mathematical reasoning domains. Following the methodology established in Shen et al. (2024), we carefully curate datasets across three critical reasoning domains with strategic subset selection to ensure balanced coverage.

Table 3: Real-World Training Corpus for Reasoning Capability Evaluation. The dataset contains totally around 2.3 billion tokens, which comes from three categories: natural language, code and math data.

| Category | Dataset | Tokens (M) |
|---|---|---|
| Natural Language SFT Data | xP3x | 317 |
| | OpenAssistant | 20.9 |
| | OpenHermes | 396 |
| | UltraChat | 251 |
| Code SFT Data | Magicoder-OSS | 50.5 |
| | Magicoder-Evol | 74.72 |
| | Code-290k-ShareGPT | 159.99 |
| | CommitPackFT | 38 |
| | Evol-Code Alpaca | 74.77 |
| Math Data | Open-web-math | 317 |
| | algebraic-stack | 353.4 |
| | TemplateGSM | 79.7 |
| | StackMathQA | 225 |
| | OpenR1-Math-220k | 46.7 |

**Data Example**  We show some training data examples for each categories below:

(1) Natural Language (example from OpenAssistant dataset):

**Question:** How can one fight back when a monospony had been created?

**Answer:** Monopsony refers to a market structure where there is only one buyer of a good or service. In the context of labor markets, a monopsony occurs when there is only one employer in a particular geographic area or industry, giving them significant power to set wages and working conditions. Research has shown that the existence of monopsonies can lead to lower wages and reduced job opportunities for workers. For example, a study by the National Bureau of Economic Research found that in industries with high levels of concentration, workers earn lower wages and are less likely to receive benefits such as health insurance.

(2) Code (example from Magicoder-OSS dataset):

**Problem:** Write a Python function to find the longest palindromic substring.

**Solution:**
```python
def longest_palindrome(s:  str) -> str:
  if not s:  return ""

  def expand_around_center(left, right):
    while left >= 0 and right < len(s) and s[left] == s[right]:
      left -= 1; right += 1
      return right - left - 1

  start, max_len = 0, 0
  for i in range(len(s)):
    len1 = expand_around_center(i, i)     # odd-length
    len2 = expand_around_center(i, i + 1) # even-length
    current_max = max(len1, len2)
    if current_max > max_len:
      max_len = current_max
      start = i - (current_max - 1) // 2
  return s[start:start + max_len]
```

(3) Math (example from Open-web-math dataset):

---

**Text:** Physical Quantity Analogous to Inductance

1. **May 12, 2013** — `tapan_ydv`

   Hi,

   I understand that some physical quantities in electromagnetism are analogous to physical quantities in heat transfer. For instance, electric field is analogous to temperature gradient.

   I want to know which physical quantity in heat transfer is analogous to inductance ($L$)?

   Regards,

2. **May 12, 2013** — `tiny-tim`

   welcome to pf!

   hi tapan_ydv! welcome to pf!

   i don't know about a heat transfer analogy,

   but a hydraulics analogy is a paddle-wheel.

   A heavy paddle wheel placed in the current. The mass of the wheel and the size of the blades restrict the water's ability to rapidly change its rate of flow (current) through the wheel due to the effects of inertia, but, given time, a constant flowing stream will pass mostly unimpeded through the wheel, as it turns at the same speed as the water flow …

   (from   `http://en.wikipedia.org/wiki/Hydraulic_analogy#Component_equivalents`)

3. **May 12, 2013** — `technician`

   In mechanics … inertia.

4. **May 12, 2013** — `tiny-tim`

   how?

5. **May 12, 2013** — `technician`

   Reluctance to change … as in a paddle wheel.

*Last edited: May 12, 2013*

---

# F   MODEL ARCHITECTURES AND TRAINING DETAILS

## F.1   GPT-2 MODEL CONFIGURATIONS FOR KNOWLEDGE CAPACITY EXPERIMENTS

We established four GPT-2 models with increasing parameter counts while maintaining architectural consistency. The models are configured as follows:

- **5M parameters**: 4 layers, hidden size 92
- **10M parameters**: 4 layers, hidden size 128
- **15M parameters**: 4 layers, hidden size 156
- **20M parameters**: 4 layers, hidden size 184

For VLD experiments, we applied the three patterns (Sequence, Cycle, Inverse Cycle) to the 5M and 20M baseline models with repetition factors $\times\{1, 2, 3\}$.

**Training Configuration:** we show the training configurations in Table 4.

## F.2   GPT-2 MODEL CONFIGURATIONS FOR REASONING EXPERIMENTS

We constructed four GPT-2 based models with identical per-layer parameter counts but varying native layer depths:

- **4 layers** ( 50M parameters): Base architecture for VLD application
- **8 layers** ( 100M parameters): Base architecture for VLD application
- **12 layers** ( 150M parameters): Baseline comparison only
- **16 layers** ( 200M parameters): Baseline comparison only

Table 4: Training Hyperparameters and Configuration for Knowledge Capacity Experiments

| Parameter | Value |
|---|---|
| Optimizer | Adam |
| Learning rate | $2 \times 10^{-5}$ |
| Precision | FP16 |
| Micro-batch size | 24 |
| Sequence length (training) | 768 |
| Sequence length (evaluation) | 1024 |
| Max position embeddings | 1024 |
| Tensor model parallel size | 1 |
| Pipeline model parallel size | 1 |
| Seed | 1234 |
| Tokenizer type | RandomNumberTokenizer |
| Hardware | $8 \times$ A100 GPUs |

VLD patterns were applied to 4-layer and 8-layer models with multiplication factors $\times\{1, 2, 3\}$ (and up to $\times 5$ for cycle pattern).

**Training Configuration:** we show the training configurations in Table 5.

Table 5: GPT-2 Training Configuration for Reasoning Experiments

| Parameter | Value |
|---|---|
| **Model Architecture** | |
| Number of layers | 4 (base model) |
| Hidden size | 768 |
| Number of attention heads | 12 |
| Sequence length | 768 |
| Max position embeddings | 2048 |
| Attention backend | Auto |
| **Training Hyperparameters** | |
| Micro batch size | 48 |
| Global batch size | 768 |
| Training iterations | 30,000 |
| Learning rate | 0.0005 |
| LR decay style | Cosine |
| LR warmup fraction | 0.019 |
| Weight decay | 0.001 |
| Adam beta1 | 0.9 |
| Adam beta2 | 0.95 |
| Gradient clipping | 1.0 |
| Precision | FP16 |

## F.3 LLaMA-3.2-3B-INSTRUCT FINE-TUNING SETUP

For real-world evaluation, we fine-tune both Base and Cycle VLD variants of LLaMA-3.2-3B-Instruct on the multi-domain corpus. Both models are initialized with identical pretrained weights to ensure fair comparison, with the Cycle VLD variant incorporating the layer repetition pattern before training commences. The training details are listed in Table 6.

Table 6: LLaMA-3.2-3B-Instruct Fine-tuning Configuration

| Parameter | Value |
|---|---|
| Base model | LLaMA-3.2-3B-Instruct |
| Total parameters | 3,218,828,288 |
| Trainable parameters (LoRA) | 6,078,464 (0.19%) |
| Fine-tuning method | LoRA |
| LoRA target modules | gate_proj, v_proj, k_proj, up_proj, o_proj, down_proj, q_proj |
| Optimizer | AdamW |
| Precision | BFloat16 |
| DeepSpeed stage | ZeRO Stage 3 |
| World size | 8 GPUs |
| Train batch size | 192 |
| Micro batch size per GPU | 6 |
| Gradient accumulation steps | 4 |
| Gradient clipping | 1.0 |
| Distributed training | DeepSpeed ZeRO-3 + LoRA |
| Parameter offloading | Enabled |
| Gradient checkpointing | Enabled |
| Memory optimization | ZeRO-3 with parameter persistence |

## G    EVALUATION PROTOCOLS AND METRICS

### G.1    KNOWLEDGE CAPACITY MEASUREMENT METHODOLOGY

After training, we input the training sequence to the LLM and compute the sum of entropy of the model's softmax outputs, then compare the total entropy of the dataset with this measured value. The difference, $\Delta H$, represents the amount of information successfully absorbed by the model.

The mean absorbed information entropy is defined as the average amount of information absorbed per token, calculated as the reduction from the original uncertainty (15.6 bits for $n = 50257$) to what the model predicts.

### G.2    REASONING CAPABILITY ASSESSMENT PROTOCOLS

**In-Distribution Evaluation:**  We test models on 500 synthetic iGSM samples with exactly 15 operations, matching the maximum complexity seen during training. Each problem requires multi-step mathematical reasoning, and we calculate exact-answer accuracy by comparing model-generated solutions against ground truth labels using string matching on the final numerical answers.

**Out-of-Distribution Evaluation:**  We assess model performance on higher complexity problems containing 20 and 21 operations, respectively. These problems maintain the same mathematical reasoning structure but require additional computational steps. All evaluations use identical decoding parameters (greedy decoding) and prompting strategies across model variants to ensure fair comparison.

### G.3    REAL-WORLD BENCHMARK EVALUATION DETAILS

We conduct comprehensive evaluations on diverse real-world reasoning benchmarks following established evaluation protocols (Habib et al., 2023):

- **Mathematical reasoning**: Math500 (Hendrycks et al., 2021) and AIME (Mathematical Association of America)

- **Scientific reasoning**: GPQA (Rein et al., 2024)

- **Code generation**: HumanEval (Chen et al., 2021) and MBPP (Austin et al., 2021)

For evaluation metrics, we maintain consistency with prior work (Habib et al., 2023; Hendrycks et al., 2021; Rein et al., 2024), employing exact match accuracy for mathematical and scientific reasoning tasks (Math500, AIME, GPQA) and pass@1 accuracy for code generation benchmarks (HumanEval, MBPP) to measure functional correctness.

**Test data examples**    We show some test data examples for each categories below:

HumanEval:

---

**Problem:**

```python
from typing import List

def has_close_elements(numbers:  List[float], threshold:  float) -> bool:
  """ Check if in given list of numbers, any two numbers are closer
  than the given threshold.
  >>> has_close_elements([1.0, 2.0, 3.0], 0.5)
  False
  >>> has_close_elements([1.0, 2.8, 3.0, 4.0, 5.0, 2.0], 0.3)
  True
  """
```

**Solution:**

```python
def has_close_elements(numbers:  List[float], threshold:  float) -> bool:
  for idx, elem in enumerate(numbers):
    for idx2, elem2 in enumerate(numbers):
      if idx != idx2:
        distance = abs(elem - elem2)
          if distance < threshold:
            return True

  return False
```

---

MBPP:

---

**Problem:** Write a function to find the longest chain which can be formed from the given set of pairs.

**Solution:**

```python
class Pair(object):
  def __init__(self, a, b):
    self.a = a
    self.b = b

def max_chain_length(arr, n):
  best = 0
  mcl = [1 for _ in range(n)]
  for i in range(1, n):
    for j in range(0, i):
      if arr[i].a > arr[j].b and mcl[i] < mcl[j] + 1:
        mcl[i] = mcl[j] + 1
  for i in range(n):
    if best < mcl[i]:
      best = mcl[i]
  return best
```

---

Math500:

**Problem:** If $f(x) = \frac{3x-2}{x-2}$, what is the value of $f(-2) + f(-1) + f(0)$? Express your answer as a common fraction.

**Solution:** $f(-2) + f(-1) + f(0) = \frac{3(-2)-2}{-2-2} + \frac{3(-1)-2}{-1-2} + \frac{3(0)-2}{0-2} = \frac{-8}{-4} + \frac{-5}{-3} + \frac{-2}{-2} = 2 + \frac{5}{3} + 1 = \boxed{\frac{14}{3}}$

AIME:

**Problem:** Let $x, y$ and $z$ be positive real numbers that satisfy the following system of equations:

$$\log_2\left(\frac{x}{yz}\right) = \frac{1}{2}$$

$$\log_2\left(\frac{y}{xz}\right) = \frac{1}{3}$$

$$\log_2\left(\frac{z}{xy}\right) = \frac{1}{4}$$

Then the value of $\left|\log_2(x^4 y^3 z^2)\right|$ is $\frac{m}{n}$ where $m$ and $n$ are relatively prime positive integers. Find $m + n$.

**Solution:** Denote $\log_2(x) = a$, $\log_2(y) = b$, and $\log_2(z) = c$.
Then, we have: $a - b - c = \frac{1}{2}$, $-a + b - c = \frac{1}{3}$, $-a - b + c = \frac{1}{4}$.
Now, we can solve to get $a = \frac{-7}{24}$, $b = \frac{-9}{24}$, $c = \frac{-5}{12}$. Plugging these values in, we obtain $|4a + 3b + 2c| = \frac{25}{8} \implies \boxed{033}$.

GPQA:

**Problem:** Two quantum states with energies $E_1$ and $E_2$ have lifetimes of $10^{-9}$ s and $10^{-8}$ s, respectively. We want to clearly distinguish these two energy levels. Which of the following options could be their energy difference so that they can be clearly resolved?
**Solution:** $10^{-4}$ eV

# H    ADDITIONAL EXPERIMENTAL RESULTS

## H.1    LAYER SELECTION STRATEGY ANALYSIS

We also conducted ablation studies to assess how the placement of shared layers affects performance. Using an 8-layer model with cycle pattern, we compared different layer selection strategies: applying VLD to all layers (1-8), early layers only (1-4), and later layers only (5-8).

Table 7: GSM Performance of the 8-Layer Model with Varying Layer Selection Strategies

| VLD Pattern | No VLD | 1-8 layers | 1-4 layers | 5-8 layers |
|---|---|---|---|---|
| cycle_8 | 60.5 | 62.0 | 54.2 | 64.2 |

# I    LIMITATIONS AND FUTURE WORK

While our comprehensive experiments demonstrate VLD's effectiveness across multiple configurations, several limitations constrain the scope and generalizability of our findings. The non-monotonic scaling behavior observed in our experiments, where performance occasionally decreases at higher VLD factors, warrants deeper investigation to understand the underlying mechanisms and identify potential fixed points where further scaling yields diminishing returns.

The non-monotonic behavior presents intriguing research opportunities: systematically exploring whether VLD has inherent fixed points beyond which performance no longer improves, investigating optimal repetition patterns through combination with other techniques like mixture-of-experts or

sparse attention mechanisms, and developing principled methods to identify the best VLD configuration for specific tasks and model scales.

