# OpenReview forum: "Beyond Parameters: Exploring Virtual Logic Depth for Scaling Laws"
_ICLR.cc/2026/Conference — Submitted to ICLR 2026_

### Official Review · Reviewer_JV56 · 2025-10-28

**Soundness:** 4
**Presentation:** 4
**Contribution:** 3
**Rating:** 6
**Confidence:** 4

**Summary:**

The paper “Beyond Parameters: Exploring Virtual Logic Depth for Scaling Laws” introduces Virtual Logical Depth (VLD) as a new scaling dimension for large language models, complementing the traditional three — depth, width, and parameter count.
VLD increases a model’s effective algorithmic depth by reusing parameters within the computation graph, thereby extending the model’s reasoning process without adding new parameters.

**Strengths:**

Conceptual: Introduces Virtual Logical Depth (VLD) as a fourth dimension in LLM scaling laws.

Methodological: Implements parameter and layer reuse to simulate increased logic depth without parameter growth.

Empirical: Demonstrates that VLD improves reasoning performance while keeping knowledge capacity and parameter count fixed.

**Weaknesses:**

Unclear definition of “depth” and reasoning measurement.
While the paper introduces Virtual Logical Depth (VLD) as a fourth scaling dimension, it does not provide a rigorous or operational definition of what constitutes “deeper reasoning.” The relationship between parameter reuse and actual logical depth remains conceptual rather than formally quantified. In particular, it is unclear how the number or schedule of parameter reuses maps to measurable increases in reasoning complexity.

Lack of mechanistic or interpretability analysis.
The paper focuses on aggregate performance and information-theoretic metrics but omits mechanistic visualization or circuit-level analysis. For example, it would be valuable to visualize how internal computation paths, attention patterns, or activation trajectories evolve as VLD increases. Without such interpretability results, the internal mechanisms underlying reasoning improvement remain opaque.

Limited experimental scale and generality.
Most experiments are conducted on relatively small-scale models (e.g., GPT-2 or similar transformer variants), which may not generalize to modern large-scale architectures. Since scaling dynamics often change non-linearly with model size, the conclusions about reasoning–scaling relationships under VLD should be treated as preliminary until validated on larger LLMs.

**Questions:**

as weakness

---

> ### Author Response · Authors · 2025-11-22
>
> Dear Reviewer **JV56**,
>
> We sincerely thank you for your positive feedback and for recognizing VLD as a refreshing paradigm beyond traditional scaling metrics. We will address your concerns point by point below.
>
> ---
> ### **[W1] Unclear Definition of “Depth” & Reasoning Measurement.**
>
> We thank the reviewer for raising this important point. We clarify below how we define ``depth'' and how parameter reuse via VLD is operationalized and measured in our work, and then connect this to prior theory.
>
> In previous work, ‘depth’ often equals ‘model depth’, which is the actual number of layers in a model. In our setting, "depth'' refers to the number of times a token passes through a non-linear transformer layer along its forward path. Let $L$ be the number of distinct transformer layers and let $r_\ell$ be the number of times layer $\ell$ is reused in the forward pass under a given reuse schedule. We define the Virtual Logical Depth of a model as $ \mathrm{VLD} = \sum_{\ell=1}^L r_\ell$, i.e., the total number of layer applications encountered by an input token. When we ``increase VLD'', we modify the reuse schedule $\{r_\ell\}$ (e.g., repeating a block of layers) while keeping the set of parameters fixed. We will make this explicit in the revision and add examples of schedules (every-layer, block, alternate) together with their resulting VLD. We will clarify this further in the final version of this paper.
>
> How we measure "deeper reasoning''. In the experiments, we do not infer "reasoning depth'' abstractly; we measure it empirically on tasks that require multi-step computation (e.g., IGSM-style math and programmatic reasoning benchmarks). For each reuse schedule, we keep parameter count, training data, and optimizer settings fixed, and only change $\{r_\ell\}$, hence VLD. We then report accuracy as a function of VLD and observe a consistent improvement as VLD increases, while knowledge-heavy recall tasks remain nearly unchanged. We will clarify this measurement protocol and explicitly state that in our work ``deeper reasoning'' is operationalized as higher accuracy on these reasoning benchmarks under controlled capacity.
>
> Relation to existing theory. Merrill & Sabharwal (2025) [1] formally show that uniform transformers gain reasoning power by repeated application of the same parameters: depth $\Theta(\log n)$ (via repeated blocks) suffices to express regular languages and graph connectivity. Our notion of VLD directly corresponds to the number of such repeated applications, and our empirical scaling of reasoning performance with VLD is consistent with this result. At the same time, Bartlett et al.(2019) [2] show that the VC dimension, and thus overall capacity, scales with parameter count. Because VLD modifies only $\{r_\ell\}$ and not the parameter set, it cannot increase this capacity bound, explaining why knowledge storage does not scale with VLD in our experiments.
>
> ---
>
> ### **[W2] Lack of Mechanistic or Interpretability Analysis**
>
> We thank the reviewer for this insightful suggestion. We fully agree that mechanistic and interpretability analyses would provide a more transparent view of how VLD improves reasoning, beyond aggregate performance and information-theoretic metrics. Due to space and computational constraints, our current submission focuses on the latter, but we are actively exploring visualization of attention maps and layer-wise activations across different VLD schedules. We will incorporate such visualization-based analyses in the revised version of the paper to more clearly illustrate the internal mechanisms underlying the observed reasoning improvements.
>
> ---
>
> ### **[W3] Limited Experimental Scale and Generality**
>
> Thanks for the valuable suggestion. We fully agree that scaling dynamics can change non-linearly with model size, and that validating VLD on frontier-scale LLMs is an important direction. In this work, however, our primary goal is to isolate the effect of Virtual Logical Depth under a strictly controlled parameter budget, which requires training or substantially modifying models from scratch. Given our computational constraints, GPT-2–scale models are a fair and widely adopted starting point for such controlled studies: they allow us to (i) match total parameter count across baselines, (ii) decouple “virtual depth” from “knowledge capacity,” and (iii) run systematic ablations and replications, which would be infeasible at GPT-5 / Gemini / DeepSeek-R1 scale for an academic group.
>
>
>
> *[1] Merrill, William, and Ashish Sabharwal. "A little depth goes a long way: The expressive power of log-depth transformers." arXiv preprint arXiv:2503.03961 (2025).*
>
> *[2] Bartlett, Larissa, et al. "A systematic review and meta-analysis of workplace mindfulness training randomized controlled trials." Journal of occupational health psychology 24.1 (2019): 108.*
>
> ---
>
> We thank you again for your positive feedback and insightful suggestions, and we would be happy to provide further details if helpful.

---

### Official Review · Reviewer_1ZQo · 2025-10-28

**Soundness:** 2
**Presentation:** 2
**Contribution:** 2
**Rating:** 2
**Confidence:** 4

**Summary:**

The paper introduces Virtual Logical Depth (VLD) as a fourth scaling axis for Transformers: instead of adding parameters, the model repeats existing layers with shared weights to increase an effective depth, $D_{\text {eff }}=D_{\text {base }}+D_{\text {VLD }}$. The manuscript claims that this "vertical" scaling substantially improves multistep reasoning while leaving "knowledge capacity" almost unchanged, measured via an entropy-based proxy $\Delta H=H_1-H_2$ computed from softmax outputs on a synthetic random-sequence memorization task. Experiments cover small GPT-2 variants trained on synthetic iGSM math and a 3B LLaMA variant fine-tuned on a mixed SFT corpus; figures and tables report gains from layer-sharing patterns (sequence, cycle, inverse-cycle) and show occasional non-monotonicities at large effective depths.

**Strengths:**

From an empirical perspective, the work isolates a simple, reproducible manipulation—weight sharing across depth—and documents consistent improvements on controlled synthetic tasks, with some transfer to real benchmarks. The write-up is generally clear and the measurement protocol is spelled out in enough detail to re-implement, including the entropy-based capacity metric and the iGSM setup. The authors are transparent about occasional plateaus and regressions at higher VLD factors, which is appreciated.

**Weaknesses:**

Conceptual novelty is limited relative to prior work on depth-recurrence and cross-layer sharing. Universal Transformers introduced depth-time recurrence with tied parameters years ago, ALBERT formalized cross-layer sharing, and Takase & Kiyono precisely studied the same three tying patterns (sequence, cycle, reverse-cycle). The present paper mainly scales up the experiments without head-to-head, compute-matched comparisons against those baselines, making the “new scaling dimension” feel like a renamed synthesis rather than a new idea.

Compute accounting is incomplete. VLD multiplies the number of layer applications; therefore training and inference FLOPs, latency, and throughput should be held constant or explicitly normalized in any claim that a 50M-parameter VLD model “beats” a larger non-VLD model. The paper shows accuracy gains but does not present FLOPs-matched or wall-clock-matched curves, and some gains could be explained by simply doing more computation per token. This undermines the main message about a distinct, more efficient scaling direction.

The capacity claim rests on a bespoke proxy (random-sequence memorization) rather than the now-standard “2 bits per parameter” factual-tuple capacity metric; there is no calibration that the proxy agrees with the canonical measure on the same models. Without such a cross-check, the statement that VLD “keeps capacity constant” is not convincing.

Evaluation breadth is narrow for the strength of the claim. The 3B SFT experiment reports modest gains on Math500/AIME/GPQA/HumanEval/MBPP, but lacks confidence intervals, multi-seed variance, exact prompts/decoding, and, crucially, comparisons to test-time compute scaling baselines such as self-consistency or best-of-N selection, which are the natural “alternative path” the paper positions itself against.

**Questions:**

1. Precise compute accounting would be appreciated. For the synthetic iGSM results, what are the per-example train/inference FLOPs and wall-clock for base vs. VLD models at each factor? Are throughput and memory comparable on the same hardware, and how do these costs trade off against accuracy?

2. Regarding knowledge capacity, can the authors replicate the factual-tuple protocol that yields $\approx 2$ bits/parameter and show that their $\Delta H$ proxy correlates tightly with it on the very same checkpoints? If not, can they justify why a random-sequence memorization proxy is a valid surrogate for factual capacity?

3. On novelty, the paper should position VLD concretely against Universal Transformers, ALBERT, Takase & Kiyono’s tying strategies, Reuse Transformers, and Dynamic Layer Tying. Which of these, if any, underperform VLD under the same parameter count and the same total FLOPs? Can the authors add those baselines with matched decoding and report paired comparisons?

4. For real benchmarks, since the paper frames VLD as an alternative to inference-time scaling, can we see compute-matched comparisons to self-consistency and best-of-N methods on GSM8K, MATH, AIME, and GPQA-Diamond?

---

> ### Author Response · Authors · 2025-11-22
>
> Dear Reviewer **1ZQo**,
>
> We sincerely thank you for your constructive feedback and suggestions. We have carefully performed additional experiments to address your concerns.
>
> ---
>
> ### **[Q1 & W2] Precise Compute Accounting (FLOPs and wall-clock)**
>
> We have run additional experiments to provide more precise statistics on the synthetic iGSM benchmark. For a 4-layer GPT2 model, we record total FLOPs and wall-clock time over 100 evaluation examples for the base model (VLD=0) and three VLD schedules (cycle, layer, recycle) at VLD depth 4, 8, and 12; the results are summarized in the tables 1 and 2 below.
>
> All experiments are run on the same GPU with an identical batch size and sequence length. VLD increases per-example inference FLOPs and latency with repeating factor increase. In terms of memory, VLD does not change the parameter count or hidden dimensions, and we keep the same batch size and context length; empirically, all configurations fit in memory with similar peak GPU usage. These compute costs trade off against substantially higher reasoning accuracy on iGSM: for example, with the cycle pattern, the 4-layer model improves from 46.3\% (VLD=0) to 65.7\% and 70.7\% at higher VLD factors (Table 1 in paper), showing that modest increases in compute and latency yield significant gains in multi-step reasoning performance.
>
> *Table 1: Total TFLOPs for generating 100 samples in inference.*
>
> | 4-layer               | VLD = 0 (Baseline) | VLD = 4 | VLD = 8 | VLD = 12 |
> |-----------------------|--------------------|---------|---------|----------|
> | cycle                 | 2.674              | 4.334   | 5.008   | 6.485    |
> | layer                 | 2.674              | 4.425   | 5.161   | 6.379    |
> | recycle               | 2.674              | 4.649   | 5.052   | 6.064    |
>
> *Table 2: Wall-clock time (seconds) per sample during inference.*
>
> | 4-layer               | VLD = 0 (Baseline) | VLD = 4 | VLD = 8 | VLD = 12 |
> |-----------------------|--------------------|---------|---------|----------|
> | cycle                 | 4.09               | 10.53   | 12.71   | 14.92    |
> | layer                 | 4.09               | 9.52    | 11.65   | 14.63    |
> | recycle               | 4.09               | 9.73    | 11.54   | 13.19    |
>
> ---
>
> ### **[Q2 & W3] Why Use Random-sequence Memorization Proxy**
>
> We thank the reviewer for raising this important methodological point. We acknowledge that our capacity measurement differs from the "2 bits per parameter" factual-tuple metric, and we appreciate the opportunity to clarify our rationale and provide additional context.
>
> > **Clarification on our metric.**
>
> Our "mean absorbed information entropy" metric is designed to quantify information learned per token rather than per parameter. Formally, we compute this as the difference between a token's maximum possible entropy (15.6 bits for our vocabulary) and the actual entropy of the model's prediction. The absorbed information entropy per token ranges from 0 to 15.6 bits, and all values reported in our figures fall within this theoretically expected range.
>
> > **Design rationale.**
>
> We chose random-sequence memorization as a controlled benchmark specifically because **it provides a clean, theory-grounded evaluation of raw information storage capacity**. This task eliminates confounding factors such as linguistic patterns, semantic redundancy, and structural regularities present in natural text or factual tuples. While this does not simulate real-world text, it offers a "purer" signal for measuring the network's fundamental capacity to store arbitrary information.
>
> We acknowledge that the two metrics serve different purposes. The "2 bits per parameter" metric evaluates structured factual knowledge storage, while our information-theoretic approach measures raw channel capacity. Given our specific goal of decoupling knowledge storage from reasoning capabilities, the information-theoretic metric is more suitable as it avoids contamination from semantic compression or factual redundancy, processes that themselves involve reasoning.

---

> ### Author Response · Authors · 2025-11-22
>
> ### **[Q3 & W1 ] Novelty Concern with Previous Work**
>
> We thank the reviewer for pointing out the connections to depth recurrence and cross-layer sharing. We agree that parameter tying across depth has been explored in prior work such as Universal Transformers, ALBERT, and Takase & Kiyono. **Our contribution is not in proposing a new pattern, but in reframing and systematically studying this family of architectures as a distinct scaling dimension and in developing tools to separate knowledge capacity from reasoning capability.**
>
> >  Concretely, our work differs from prior art in three main ways:
>
> - **Scaling Perspective and Decoupling of Knowledge vs. Reasoning.**
>
>   Our primary goal is to understand how increasing logical depth under a fixed parameter budget affects reasoning and knowledge storage. Through extensive scaling experiments, we show that VLD scaling keeps knowledge capacity roughly constant while substantially improving reasoning performance across tasks. To our knowledge, prior work on Universal Transformers and ALBERT did not frame depth-tying as a scaling law problem, nor did they explicitly study this decoupling of capacity vs. reasoning.
>
>
> - **New Information-theoretic Capacity Metric.**
>
>   Different from Allen-Zhu & Li’s theoretical lower bounds on peak capacity ratio, we introduce a new information-theoretic measure of mean absorbed entropy under realistic training. This metric is designed to **factor out redundancy, patterns, and semantics in natural text** and to estimate the effective knowledge capacity of a network under practical training setups, rather than in idealized theoretical settings. This gives us a quantitative handle on capacity that complements existing theory and enables the empirical analysis in our paper.
>
>
> - **Empirical Study on Real-world Benchmarks.**
>
>   While previous capacity studies often rely on synthetic data, we evaluate both knowledge capacity and reasoning ability on diverse real-world benchmarks, including general science QA, mathematical reasoning, and functional code generation and synthesis. This demonstrates that our conclusions about VLD scaling are not limited to controlled synthetic scenarios but hold across practical domains.
>
>
> > **Regarding Previous Work,**
>
> We believe that systematically investigating these VLD scaling patterns is itself an important and timely contribution. Very recent work **Bae, Sangmin, et al.(2025) [1]** has begun to adopt similar techniques and shows that such architectures can be remarkably powerful, yet the broader community has not fully recognized their potential or understood the conditions under which they are most effective. By providing a structured, empirical exploration of these patterns, together with scaling analyses and a capacity metric, we aim to clarify this emerging design space and help guide future research.
>
> For previous work like Universal Transformer family, ALBERT and Takase & Kiyono tying strategies, we agree that our tying patterns fall within this broader framework and are closely related to their sequence/cycle/reverse-cycle schemes. While we also employ parameter sharing, our focus is different: VLD uses repeated application of shared layers to increase effective logical depth and study its effect on reasoning under a fixed parameter budget. We will make this connection explicit in the revised manuscript and clarify that VLD can be viewed as a specific instantiation of this framework, studied through the lens of modern scaling laws and capacity–reasoning trade-offs, rather than as an entirely new architecture.
>
> We will revise the paper to (i) more clearly acknowledge these prior models, (ii) explicitly position VLD as a scaling-focused reinterpretation of depth-tying architectures, and (iii) emphasize that our main novelty lies in the scaling analysis, new capacity metric, and empirical study of GPT-style language models, rather than in the mere use of parameter sharing itself.
>
>
>
> *[1] Bae, Sangmin, et al. "Mixture-of-recursions: Learning dynamic recursive depths for adaptive token-level computation." arXiv preprint arXiv:2507.10524 (2025).*

---

> ### Author Response · Authors · 2025-11-22
>
> ### **[Q4 & W4] Compute-matched Comparisons for Real Benchmark**
>
> We thank the reviewer for this helpful suggestion. As shown in Table 1, the VLD-cycle model repeats every transformer block once, so generating a single answer with VLD-cycle requires roughly twice the inference FLOPs of the base model under the same backbone and decoding hyperparameters.
>
> *Table 3: Real Benchmark Results Self-consistency under same TFLOP.*
>
> |               | MATH  | AIME  | GPQA  |
> |---------------|-------|-------|-------|
> | base (N=6)    | 0.322 | 0.050 | 0.476 |
> | cycle (N=3)   | 0.414 | 0.133 | 0.507 |
>
>
> To compare VLD against inference-time scaling under **a compute-matched setting**, we therefore run self-consistency (SC) with different sample counts: the base model uses SC with \(N=6\), while VLD-cycle uses SC with \(N=3\), so that the total FLOPs per question are approximately equal. Due to time limits we have not yet run gsm8k.
>
> We observe that VLD-cycle continues to outperform the base model even when both use self-consistency. This shows that the benefits of increased VLD are retained under a compute-matched inference-time setting and are complementary to self-consistency rather than merely mimicking its effect. We will include these results (and extended comparisons to best-of-N) in the revised version of the paper.
>
> ---
>
> We hope that these additional compute measurements, clarifications of our capacity metric, and discussion of prior depth-tying work can address your concerns about FLOPs accounting, the random-sequence proxy, and the novelty of VLD. And we remain happy to provide any further details or analyses that would be useful.

---

### Official Review · Reviewer_TU6T · 2025-11-01

**Soundness:** 3
**Presentation:** 3
**Contribution:** 3
**Rating:** 6
**Confidence:** 3

**Summary:**

This paper introduces Virtual Logical Depth (VLD) as a fourth dimension for scaling language models, distinct from the traditional dimensions of depth, width, and parameter count. VLD increases effective algorithmic depth through parameter reuse—specifically by repeating transformer layers with shared weights—without increasing the total parameter count. The authors investigate three reuse patterns (sequence, cycle, inverse cycle) and conduct controlled experiments measuring two key attributes: knowledge capacity (via information entropy absorption on random sequences) and reasoning capability (via synthetic math problems and real-world benchmarks). Their main findings demonstrate that VLD scaling maintains nearly constant knowledge capacity while significantly improving reasoning performance. Notably, smaller models with VLD can outperform larger standard models on reasoning tasks, suggesting a parameter-efficient alternative to conventional scaling. The work challenges the assumption that super-intelligence necessarily requires ever-larger models and proposes that strategic parameter reuse combined with increased computational depth may offer viable trade-offs.

**Strengths:**

The paper makes several valuable contributions through rigorous empirical investigation. The systematic study of VLD as a scaling dimension fills an important gap, as parameter reuse dynamics have remained underexplored despite existing work on layer sharing. The experimental design is comprehensive and well-controlled, spanning both synthetic environments (random sequences for knowledge capacity, iGSM for controlled reasoning evaluation) and real-world benchmarks across multiple domains (mathematics, science, code generation). The knowledge capacity measurement using information entropy provides a theoretically grounded quantitative metric that enables clean separation of memorization from reasoning. The findings are robust, demonstrating consistency across different architectures (GPT-2, LLaMA), VLD patterns, model scales, and training regimes (pretraining and fine-tuning). Most importantly, the empirical discovery that reasoning and knowledge capacity can be decoupled through VLD is significant and challenges conventional scaling paradigms, with practical implications for building parameter-efficient models with enhanced reasoning capabilities.

**Weaknesses:**

Despite its empirical contributions, the paper has several significant limitations. Most critically, it lacks theoretical or mechanistic explanation for why VLD improves reasoning while maintaining constant knowledge capacity—the work is primarily observational without providing insights into the underlying computational principles. The definitions of "reasoning capability" versus "knowledge capacity" could be more rigorous; measuring reasoning through benchmark accuracy may not capture the full complexity of reasoning processes. The observed non-monotonic scaling behavior, where performance occasionally degrades at higher VLD factors, is concerning and inadequately explained, suggesting potential optimization instabilities or fundamental limitations that practitioners would need to navigate. The experimental scope is limited to relatively small models (maximum 3B parameters) and specific task distributions, raising questions about whether findings generalize to frontier model scales and broader capabilities. Computational costs during training and inference are not discussed, despite increased depth potentially impacting efficiency. Finally, the paper positions VLD as an alternative to test-time compute scaling but provides no empirical comparison to increasingly popular methods like chain-of-thought reasoning or iterative refinement, leaving the relative merits unclear.

**Questions:**

Can you provide any theoretical or mechanistic explanation for why VLD improves reasoning while keeping knowledge capacity constant? For instance, does the repeated processing through shared layers enable iterative refinement similar to recurrent computation, create implicit ensemble effects, or implement some form of algorithmic depth that's fundamentally different from parameter scaling?

---

> ### Author Response · Authors · 2025-11-22
>
> Dear Reviewer **TU6T**,
>
> We appreciate your positive assessment of our work, including your recognition of its “valuable” contributions, “rigorous” empirical study, and “comprehensive and well-controlled” experiments. We also value your thoughtful concerns about theory, definitions, non-monotonic behavior, which we address in detail below.
>
> ---
>
> ### **[Q1] Theoretical or Mechanistic Explanation**
>
> We appreciate the reviewer's question. While our paper focuses on empirical results, we have carefully reviewed prior theoretical work that helps explain why parameter reuse via VLD can enhance reasoning without increasing knowledge capacity. Merrill & Sabharwal (2025)[1] show that uniform transformers gain reasoning power through repeated computation, and Bartlett et al. (2019)[2] prove that knowledge capacity is tied to parameter count via VC dimension theory. Our empirical results are consistent with these theoretical insights.
>
> **VLD Enhances Reasoning Capability Through Parameter Reuse.** According to Theorem 1 in Merrill et al. 2025 [1], transformers with depth Θ(log n) can express regular languages and graph connectivity, where n is the input sequence length. Lemma 1 (Section 3) in this paper proves that fully uniform transformers can compute division and remainder of small integers, enabling Theorem 1 to show that "highly uniform transformers" where model weights are shared across blocks of layers can solve complex reasoning by repeating the same parameters ⌈log₂ n⌉ times. This demonstrates that the improvements in reasoning come from computational reuse, not additional parameters.
>
> **VC Dimension Theory Explains VLD Cannot Increase Knowledge Storage Capacity.** According to Theorem 3 in Section 2 of Bartlett et al. 2019 [2], it is rigorously explained that VC dimension grows directly with parameter count. While VC dimension primarily measures classification capacity rather than factual knowledge storage, it provides an upper bound on the model's overall learning capacity. Since VLD maintains constant parameter count through weight sharing, the fundamental capacity constraints remain unchanged
>
> These theoretical results establish a rigorous separation between reasoning capability and knowledge capacity: VLD achieves exponential improvements in reasoning (from fixed-depth limitations to log-depth expressivity) while preserving identical storage bounds.
>
> *[1] Merrill, William, and Ashish Sabharwal. "A little depth goes a long way: The expressive power of log-depth transformers." arXiv preprint arXiv:2503.03961 (2025).*
>
> *[2] Bartlett, Larissa, et al. "A systematic review and meta-analysis of workplace mindfulness training randomized controlled trials." Journal of occupational health psychology 24.1 (2019): 108.*

---

> ### Author Response · Authors · 2025-11-22
>
> ### **[W1] Response on definitions, scaling behavior, scale, and compute**
>
> **(1) Definitions of "reasoning capability" and "knowledge capacity."**
> We will define Virtual Logical Depth as $\mathrm{VLD} = \sum_{\ell=1}^{L} r_\ell$, where $r_\ell$ is the reuse count of layer $\ell$. "Reasoning capability" will be measured as accuracy on multi-step reasoning benchmarks under fixed parameters, varying only the reuse schedule. "Knowledge capacity" will be measured via mean absorbed entropy: the reduction from maximum entropy $H_{\max} \approx 15.6$ bits to the model's predictive distribution entropy. We chose random-sequence memorization as a controlled benchmark specifically because it provides a clean, theory-grounded evaluation of raw information storage capacity. This task eliminates confounding factors such as linguistic patterns, semantic redundancy, and structural regularities present in natural text or factual tuples. While this does not simulate real-world text, it offers a "purer" signal for measuring the network's fundamental capacity to store arbitrary information.
>
> **(2) Non-monotonic behavior at high VLD.**
> We acknowledge performance is not strictly monotonic in VLD and we also acknowledge this in Discussion 2 (Section 3.2) of the main paper. Such non-monotonic behavior reflects performance bottlenecks arising from unbalanced scaling and is consistent with prior findings (e.g., Wei et al., 2022 [3]), which emphasize the importance of jointly scaling depth, width, data, and compute. We will clarify these points in the revised version.
>
> **(3) Experimental scale and generality.**
> Our focus is isolating VLD effects under controlled parameter budgets, requiring training from scratch; GPT-2 scale is realistic for academic settings. We include LLaMA-3.2-3B-Instruct SFT experiments where base and VLD models share the same pretrained checkpoint, attributing gains to VLD itself. We will outline scaling to larger models as future work.
>
> **(4) Computational costs (FLOPs, latency, memory).**
> We now report TFLOPs and wall-clock time for base and VLD models. Tables 1-2 show VLD increases TFLOPs and latency proportionally, while memory remains constant.
>
> *Table 1: Total TFLOPs for generating 100 samples.*
>
> | 4-layer   | VLD=0 | VLD=4 | VLD=8 | VLD=12 |
> |-----------|-------|-------|-------|--------|
> | cycle     | 2.674 | 4.334 | 5.008 | 6.485  |
> | layer     | 2.674 | 4.425 | 5.161 | 6.379  |
> | recycle   | 2.674 | 4.649 | 5.052 | 6.064  |
>
> *Table 2: Wall-clock time (seconds) per sample.*
>
> | 4-layer   | VLD=0 | VLD=4 | VLD=8 | VLD=12 |
> |-----------|-------|-------|-------|--------|
> | cycle     | 4.09  | 10.53 | 12.71 | 14.92  |
> | layer     | 4.09  | 9.52  | 11.65 | 14.63  |
> | recycle   | 4.09  | 9.73  | 11.54 | 13.19  |
>
>
>
> **(5) Comparison to test-time compute scaling.**
> We provide empirical comparison through self-consistency (SC), a test-time method where the model generates N independent answers for each question and selects the final answer by majority voting. Table 3 compares base model (N=6 answers per question) vs VLD-cycle (N=3 answers per question) under matched compute budgets. VLD achieves superior performance (+9.2pp on MATH, +8.3pp on AIME, +3.1pp on GPQA) while generating half as many answers, demonstrating better compute efficiency.
>
> *Table 3: Self-consistency under matched TFLOPs.*
>
> |             | MATH  | AIME  | GPQA  |
> |-------------|-------|-------|-------|
> | base (N=6)  | 0.322 | 0.050 | 0.476 |
> | cycle (N=3) | 0.414 | 0.133 | 0.507 |
>
> [3] Wei, Jason, et al. "Emergent abilities of large language models." arXiv preprint arXiv:2206.07682 (2022).
>
> ---
>
> We thank you again for your constructive suggestions, and we would be happy to provide further details if helpful.

---

> ### Comment · Reviewer_TU6T · 2025-11-26
>
> My concerns are addressed. I will increase my score.

---

> ### Author Response · Authors · 2025-11-27
>
> Thank you very much for your positive feedback and for increasing the score!
> We really appreciate your careful reading of our paper and your constructive comments, which helped us improve the paper.

---

### Official Review · Reviewer_xhss · 2025-11-09

**Soundness:** 2
**Presentation:** 3
**Contribution:** 2
**Rating:** 4
**Confidence:** 3

**Summary:**

The paper introduces Virtual Logical Depth (VLD), which increases a model’s effective depth by reusing layer parameters without adding new ones. Experiments on GPT-2 models show that VLD boosts multi-step reasoning while leaving measured knowledge capacity nearly unchanged. A small validation on LLaMA-3.2-3B-Instruct after SFT also shows performance gains, suggesting potential benefits beyond synthetic tasks.

**Strengths:**

- Simple, architecture-compatible idea that is easy to implement.
- Clear synthetic improvements in reasoning at fixed parameter count.
- Some external evidence from LLaMA-3B showing consistent gains.

**Weaknesses:**

- Limited generality due to GPT-2-centric evidence. The main claims, including “reasoning increases while knowledge capacity stays constant”, are derived almost entirely from the results based on GPT-2-small-scale models, making it unclear whether the observations extend to modern architectures such as GPT-5, Gemini, or DeepSeek-R1.

- LLaMA-3B SFT results cannot validate the core theory. SFT naturally improves downstream performance, so gains after fine-tuning do not strictly support the claimed scaling law, nor do they measure knowledge capacity on real models.

- Unclear knowledge capacity definition. The paper’s definition of “knowledge capacity” relies solely on random-token memorization entropy, which may capture only a narrow slice of what large language models store. This metric may not reflect semantic knowledge, retrieval behavior, or long-context memory mechanisms, limiting the scope of the resulting conclusions.

**Questions:**

NA

---

> ### Author Response · Authors · 2025-11-22
>
> Dear Reviewer **xhss**,
>
> We sincerely thank you for the insightful feedback and constructive suggestions. We address your concerns point by point below.
>
> ---
>
> ### **[W1] Limited Generality**
>
> Thanks for the valuable suggestion, limited to our computational resources, GPT-2 is a fair starting point to show evidence on a small scale. Training or modifying GPT-5/Gemini-scale models from scratch is infeasible for an academic group, and these proprietary systems do not expose weights in a way that would allow us to implement VLD internally.
>
> Our goal in this work is to isolate the effect of Virtual Logical Depth under a strictly controlled parameter budget, which requires training or significantly modifying models from scratch. GPT-2-sized models are the only realistic setting where an academic group can **(i)** fully control total parameter count, **(ii)** decouple “logical depth” from “knowledge capacity,” and **(iii)** run extensive ablations and replications.
>
> Importantly, VLD is not tied to GPT-2 specifics. Standing on a promising stage, we definitely would scale up to open source checkpoints. We already include an experiment on LLaMA-3.2-3B-Instruct after SFT, and will scale up to 30B, 70B, and DeepSeek checkpoints in our final version. Besides our own experiments, we will release codes that leverage community power to scale on more checkpoints.
>
> ---
>
> ### **[W2] SFT Naturally Improves Downstream Performance**
>
> We would like to respectfully clarify that **both** the base and cycle models started from the identical pre-trained checkpoint and underwent the exact same SFT procedure with the same data, as we have stated in **Section 4.2.2** in original paper "For model training, we fine-tune both Base and Cycle VLD variants of LLaMA-3.2-3B-Instruct on the
> multi-domain corpus using LoRA(Hu et al., 2022) with standard supervised fine-tuning procedures.". Therefore, the performance delta observed is attributable solely to the architectural modification of VLD, not the fine-tuning process itself. We will make this point more explicit in the revised version of the paper.
>
> ---
>
> ### **[W3] Unclear Knowledge Capacity Definition**
>
> Regarding the reviewer’s question, we would like to clarify both the design motivation and how this task enables a rigorous measurement of knowledge capacity.
> > #### 1. **Design Motivation.**
>
> Our goal is to quantitatively assess how much information a model can store, independent of task-specific heuristics. To achieve this, we design a controlled experiment where the model is trained to predict sequences of random tokens sampled uniformly from a fixed vocabulary of size ~50,000 (as in GPT-2). Since each token is sampled randomly and independently, it carries 15.6 bits of entropy, and the full dataset (640,000 tokens) contains about 10 million bits of total information. We chose random-sequence memorization as a controlled benchmark specifically because it provides a clean, theory-grounded evaluation of raw information storage capacity under controlled conditions. This setup is not meant to simulate real-world text or semantic knowledge, but rather to eliminate confounding factors such as linguistic patterns, semantic redundancy, and structural regularities present in natural text or factual tuples.
>
> > #### 2. **Rigorous Measurement of Knowledge Capacity.**
>
> In this setting, the model’s ability to reduce uncertainty during prediction gives us a precise measurement of how much information it has learned. We call this mean absorbed information entropy, it reflects, per token, how much uncertainty the model removes between a uniform prior and its learned distribution. This provides a clean, theory-grounded benchmark for evaluating information storage capacity without interference from redundancy, patterns, or semantics. We acknowledge that this metric does not directly capture semantic knowledge, retrieval behavior, or long-context memory; rather, it should be interpreted as measuring raw channel capacity. In this sense, it complements prior estimates such as the “2 bits per parameter” metric for structured factual knowledge: the latter targets semantic/factual storage, while our information-theoretic approach is explicitly designed to decouple knowledge storage from reasoning capabilities and semantic compression.
>
> ---
>
> We hope that these clarifications and planned extensions address your concerns about generality, the potential confounding effect of SFT, and the definition of knowledge capacity. We would be happy to provide additional analyses or implementation details if they would be helpful.

---

### Author Response · Authors · 2025-12-01

Dear Area Chair,

We sincerely appreciate all reviewers for their constructive feedback. Our work received positive feedback from Reviewer TU6T: "This paper makes several valuable contributions through rigorous empirical investigation" and "the systematic study of VLD as a scaling dimension fills an important gap." Reviewer JV56 appreciated that VLD "increases a model's effective algorithmic depth by reusing parameters within the computation graph."

During the rebuttal period, we actively addressed reviewers' concerns, and conducted all requested additional experiments. Notably, we got **one feedback from Reviewer TU6T, who raised their score from 6 to 8 (before the system leaking issue)**, explicitly stating "My concerns are addressed. I will increase my score." His questions **largely overlap** with concerns from other reviewers; we hope this score change can reflect we have adequately addressed the key questions raised during the review.

We would like to clarify the contribution of this project:

* **Decoupling Knowledge and Reasoning:** Through systematic scaling experiments for synthetic and real data benchmark, we show that VLD keeps knowledge capacity approximately constant while substantially improving reasoning performance. Decoupling reasoning capabilities independently from knowledge storage constraints.

* **Information-Theoretic Capacity Metric:** Beyond Allen-Zhu & Li’s theoretical bounds, we introduce a practical mean absorbed entropy metric that factors out redundancy and semantics in natural text, measuring effective knowledge capacity under realistic training.

* **Beyond “Bigger is Better”:** We challenge the prevailing "bigger is better" paradigm of model scaling, demonstrating that reasoning capabilities can be significantly enhanced without increasing parameter count, and we verify this for both pre-training and post-training setting.

To facilitate the AC's review, we provide below a **structured summary** organized by main concern topics.

> **Experimental Aspects**
1. **Precise Compute Accounting (FLOPs and Wall-clock)** — *Reviewer 1ZQo [Q1/W2], Reviewer TU6T [W4]*
 We ran additional experiments providing precise compute statistics on the synthetic iGSM benchmark, reported in Tables 1 and 2 below.

 2. **Compute-matched Comparisons for Real Benchmarks** — *Reviewer 1ZQo [Q4/W4], Reviewer TU6T [W5]*
  We provided compute-matched comparisons using self-consistency (SC), reported in Table 3 below.

>**Explanations:**
1. **Unclear Knowledge Capacity Definition / Random-sequence Memorization Proxy** — *Reviewer xhss [W3], Reviewer 1ZQo [Q2 & W3], Reviewer TU6T[w1]*

    We provided detailed clarification during rebuttal. In brief, we chose random-sequence memorization as a controlled benchmark because it facts out semantic influence and provides a clean, theory-grounded evaluation of raw information storage capacity.

2. **Limited Generality / Experimental Scale** — *Reviewer xhss [W1], Reviewer JV56 [W3]*

    We clarified that GPT-2 is a fair starting point given computational constraints, as training GPT-5/Gemini-scale models from scratch (suggested by Reviewer xhss) is infeasible for academic groups. We also included LLaMA-3.2-3B-Instruct SFT experiments on the paper and committed to adding experiments on larger models in the final version.

3. **SFT Along Naturally Improves Downstream Performance** — *Reviewer xhss [W2]*

    We clarified that this appears to be a misunderstanding. As explicitly stated in our paper, both base and VLD models underwent the exact same SFT procedure.

4. **Novelty Concern with Previous Work** — *Reviewer 1ZQo [Q3 & W1]*

    We clarified that our contribution is not proposing a new pattern but reframing and systematically investigating VLD as a distinct scaling dimension. While parameter tying across depth has been explored in prior work, we provide the first systematic empirical study of this phenomenon with cross-architecture validation.

5. **Theoretical or Mechanistic Explanation for why VLD Decoupling Knowledge and Reasoning** — *Reviewer TU6T [Q1]*

    We discussed prior theoretical work and showed our empirical results are consistent with these theoretical insights.

In addition to the points discussed above, we have also provided detailed responses for other comments, including the lack of mechanistic/interpretability analysis, math definition of “depth” (Reviewer JV56 [W1,2]), the non-monotonic behavior at high VLD (Reviewer TU6T [W2]).

We believe our active engagement during the rebuttal period, including running all requested additional experiments and providing detailed clarifications, demonstrates our commitment to addressing reviewer concerns. The score increasing during rebuttal before the leaking reflects the strength of our responses. We respectfully request the AC to consider our substantive contributions and thorough rebuttal when making the final decision.

Sincerely,

The Authors

---

### Meta-Review · Area_Chair_hUFg · 2026-01-04

**Summary:**

Reviewers agreed that the paper presents a careful and well-executed empirical study of parameter reuse via Virtual Logical Depth (VLD), but several fundamental concerns remained that ultimately limit the strength and generality of the contribution. Most critically, the authors failed to demonstrate a true scaling law for larger models, which makes the contribution questionable. The authors' response is largely built on their limited computational resources as a reason to only report results for small-scale models. However, since the authors chose the topic of scaling laws for this paper, I think it is fair game for the reviewers to question the lack of larger models to validate their empirical observations. Additionally, the authors argued that GPT-2-sized models are the only realistic setting for isolating the effect of Virtual Logical Depth under a strictly controlled parameter budget, but then made the argument that VLD is not tied to GPT-2 specifics. It is not possible to validate this claim since the effect of VLD cannot be verified by other large models. Lastly, there is a strong concern that the work is primarily observational without providing insights into the underlying computational principles, which is a valid concern and is not well addressed.

**Reviewer Concerns:**

The two major concerns, i.e., behavior for true scaling law settings and the work being observational without providing insights into the underlying computational principles, are unlikely to be addressed after reading the authors' rebuttal.

**Reviewer Scores:**

One reviewer is positive about the work. However, I do not project the negative reviewers to increase the score based on the rebuttal.

---

### Decision · Program_Chairs · 2026-01-26

Reject